# 3D Shape Reconstruction from Vision and Touch

Edward J. Smith[1,2]*    Roberto Calandra[1]    Adriana Romero[1,2]    Georgia Gkioxari[1]

David Meger[2]    Jitendra Malik[1,3]    Michal Drozdzal[1]

[1] Facebook AI Research    [2] McGill University    [3] University of California, Berkeley

## Abstract

When a toddler is presented a new toy, their instinctual behaviour is to pick it up and inspect it with their hand and eyes in tandem, clearly searching over its surface to properly understand what they are playing with. At any instance here, touch provides high fidelity localized information while vision provides complementary global context. However, in 3D shape reconstruction, the *complementary* fusion of visual and haptic modalities remains largely unexplored. In this paper, we study this problem and present an effective chart-based approach to multi-modal shape understanding which encourages a similar fusion vision and touch information. To do so, we introduce a dataset of simulated touch and vision signals from the interaction between a robotic hand and a large array of 3D objects. Our results show that (1) leveraging both vision and touch signals consistently improves single-modality baselines; (2) our approach outperforms alternative modality fusion methods and strongly benefits from the proposed chart-based structure; (3) the reconstruction quality increases with the number of grasps provided; and (4) the touch information not only enhances the reconstruction at the touch site but also extrapolates to its local neighborhood.

## 1  Introduction

From an early age children clearly and often loudly demonstrate that they need to both look and touch any new object that has peaked their interest. The instinctual behavior of inspecting with both their eyes and hands in tandem demonstrates the importance of fusing vision and touch information for 3D object understanding. Through machine learning techniques, 3D models of both objects and environments have been built by independent leveraging a variety of perception-based sensors, such as those for vision (e.g. a single RGB image) [58, 19] and touch [72, 66]. However, vision and touch possess a clear complementary nature. On one hand, vision provides a global context for object understanding, but is hindered by occlusions introduced by the object itself and from other objects in the scene. Moreover, vision is also affected by bas-relief [39] and scale/distance ambiguities, as well as slant/tilt angles [3]. On the other hand, touch provides localized 3D shape information, including the point of contact in space as well as high spatial resolution of the shape, but fails quickly when extrapolating without global context or strong priors. Hence, combining both modalities should lead to richer information and better models for 3D understanding. An overview of 3D shape reconstruction from vision and touch is displayed in Figure 1.

Visual and haptic modalities have been combined in the literature [2] to learn multi-modal representations of the 3D world, and improve upon subsequent 3D understanding tasks such as object manipulation [41] or any-modal conditional generation [42]. Tactile information has also been used

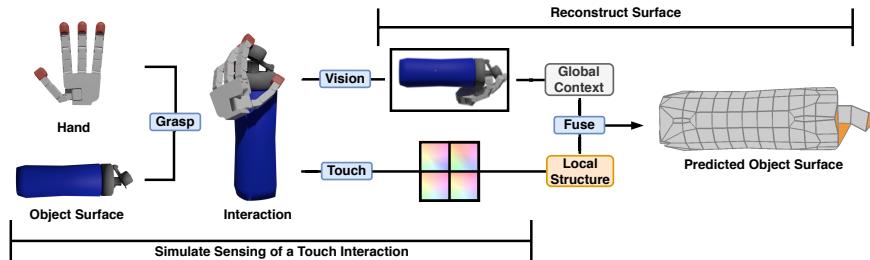

Figure 1: 3D shape understanding from vision and touch includes: (1) shape sensing with a camera and touch sensor, as well as (2) reconstruction algorithm that fuses vision and touch readings. In this paper, we introduce a dataset that captures object sensing and propose a chart-based fusion model for 3D shape prediction from multi-modal inputs. For touch, we realistically simulate an existing vision-based tactile sensor [40].

to improve 3D reconstructions in real environments. In particular, [66] leverages vision and touch sequentially, by first using vision to learn 3D object shape priors on simulated data, and subsequently using touch to refine the vision reconstructions when performing sim2real transfer. However, to the best of our knowledge, the *complementary* fusion of vision (in particular, RGB images) and touch in 3D shape reconstruction remains largely unexplored.

In this paper, we focus on this unexplored space, and present an approach that effectively fuses the global and local information provided by visual and haptic modalities to perform 3D shape reconstruction. Inspired by the papier-mâché technique of AtlasNet and its similar works [21, 71, 14] and leveraging recent advances in graph convolutional networks (GCN) [38], we aim to represent a 3D object with a collection of disjoint mesh surface elements, called *charts* [33], where some charts are reserved for tactile signals and others are used to represent visual information. More precisely, given an RGB image of an object and high spatial resolution tactile (mimicking a DIGIT tactile sensor [40]) and pose information of a grasp, the approach predicts a high fidelity local chart at each touch site and then uses the corresponding vision information to predict global charts which close the surface around them, in a fill-in-the-blank type procedure. As learning from real world robot interactions is resource and time intensive, we have designed a simulator to produce a multi-modal dataset of interactions between a robotic hand and four classes of objects, that can be used to benchmark approaches to 3D shape reconstructions from vision and touch, and help advance the field. Our dataset contains ground truth 3D objects as well as recordings from vision and tactile sensors, such as RGB images and touch readings. Results on the proposed dataset show that by combining visual and tactile cues, we are able to outperform single modality touch and vision baselines. We demonstrate the intuitive property that learning from touch exclusively translates into decreased performance, as the 3D shape reconstruction suffers from poor global context while learning from vision exclusively suffers from occlusions and leads to lower local reconstruction accuracy. However, when combining both modalities, we observe a systematic improvement, suggesting that the proposed approach effectively benefits from vision and touch signals, and surpasses alternative fusion strategies. Moreover, when increasing the number of grasps provided, we are able to further boost the 3D shape reconstruction quality. Finally, due to our model design, the touch readings not only enhance the reconstruction at the touch site but also reduce the error in the neighborhood of touch sensor position. Our main contributions can be summarized as: (1) we introduce a chart-based approach to 3D object reconstruction, leveraging GCNs to combine visual and haptic signals; (2) we build a dataset of simulated haptic object interactions to benchmark 3D shape reconstructions algorithms in this setting; and (3) through an extensive evaluation, we highlight the benefits of the proposed approach, which effectively exploits the complementarity of both modalities. Code for our system is publicly available on a GitHub repository, to ensure reproducible experimental comparison.[2]

## 2   Related Work

**3D reconstruction from vision.** There is a vast literature addressing 3D shape reconstruction from visual signals. Approaches often differ in their input visual signal – e.g. single view RGB image [59, 58, 19, 46], multi-view RGB images [12, 27, 35, 37], and depth images [54, 73] –, and their

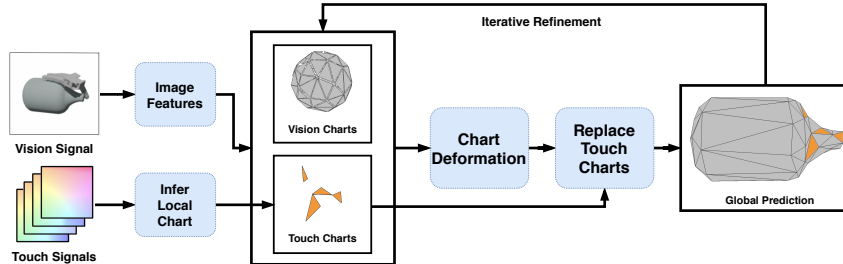

Figure 2: Our approach to 3D shape reconstruction combines a single RGB image with 4 touch readings. We start by predicting touch charts from a touch recordings, and projecting the visual signal onto all charts. Then, we feed the charts into an iterative deformation process, where we enforce touch consistency. As a result, we obtain a global prediction of deformed charts.

predicted 3D representation – e.g., orientation/3D pose [26, 17], signed distance functions [45], voxels, point clouds, and meshes [32]. Point cloud-based approaches [16, 52, 29, 47], together with voxel-based approaches [9, 59, 63, 68, 69, 70], and their computationally efficient counter-parts [53, 62, 23] have long dominated the deep learning-based 3D reconstruction literature. However, recent advances in graph neural networks [7, 13, 38, 64, 22] have enabled the effective processing and increasing use of surface meshes [36, 65, 34, 30, 25, 58, 8] and hybrid representations [20, 19]. While more complex in their encoding, mesh-based representations benefit greatly from their arbitrary resolution over other more naive representations. Our chosen representation more closely relates to the one of [20], which combines deformed sheets of points to form 3D shapes. However, unlike [20], our proposed approach exploits the neighborhood connectivity of meshes. Finally, 3D reconstruction has also been posed as a shape completion problem [59, 73], where the input is a partial point cloud obtained from depth information and the prediction is the complete version of it.

**3D reconstruction from touch.** Haptic signals have been exploited to address the shape completion problem [4, 50, 60, 48, 43]. Shape reconstruction has also been tackled from an active acquisition perspective, where successive touches are used to improve the reconstruction outcome and/or reduce the reconstruction uncertainty [5, 44, 72, 31, 15]. Most of these works use point-wise tactile sensors, while in contrast we use a high-dimensional and high-resolution sensor [40] which provide far more detailed local geometry with respect to the object being touched. In addition, these works make use only of proprioceptive and touch information, while we also tackle the problem of integrating global information from visual inputs in a principled manner. For an extensive and more general review on robotic tactile perception, we refer the reader to [43].

**3D reconstruction from vision and touch.** Many approaches exploiting vision and touch for 3D shape reconstruction rely on depth information [6, 28, 18]. In these works the depth information is represented as a sparse point cloud, augmented with touch points, which is fed to a Gaussian Process to predict implicit shape descriptors (e.g., level sets). Another line of work [66] considers RGB visual signals and uses a deep learning-based approach to produce voxelized 3D shape priors, which are subsequently refined with touch information when transferring the set-up to a real environment. Note that, following 3D shape reconstruction from touch, the previous works are concerned with the active acquisition of grasps. Moreover, [67] uses touch and partial depth maps separately to predict independent voxel models, which are then combined to produce a final prediction. In contrast to these works, we use a 4-fingered robot hand equipped with high-resolution tactile sensors – integrating such high-dimensional inputs is significantly more challenging but also potentially more useful for down-stream robot manipulation tasks.

## 3 Global and Local Reconstruction Methods

We consider the problem of 3D shape reconstruction from visual and haptic signals and leverage a deep learning approach which deforms disjoint mesh surface elements through a GCN. We assume that visual information is obtained from a single RGB image and haptic information is obtained from vision-based touch sensors with high spatial resolution, such as DIGIT [40]. More precisely, let $V$ denote the RGB image used a as vision signal. Let $T = [R_i, P_i, M_i]_{i=1}^{n_t}$ denote the touch information, where $R_i$ is one touch sensor reading, $P_i$ its corresponding position and rotation in space, $M_i$ a binary

mask indicating whether the touch is successful (i.e. the sensor is in contact with the object), and $n_t$ is the number of touch sensors. Let $O$ be the target object shape, represented as a surface mesh. The objective is to learn a function $f_\theta$ parameterized by $\theta$ that predicts an object shape reconstruction $\hat{O} = f_\theta(V, T)$ such that it best captures the surface defined by $O$. In our approach, we represent $\hat{O}$ as a set of independent surface elements, $\{C_i\}_{i=1}^{n_c}$, which we call *charts*. A chart, $C_i$, is implemented as a planar, 3D polygon mesh, composed of connected triangular faces, each defined by 3 vertex positions. Figure 3 depicts the structure of a chart, and outlines how a set of charts can be combined to form a closed 3D surface. In the left image we show how how a chart is parameterized by a simple planar mesh sheet, in the middle image, a collection of these charts, and in the right image how this collection is combined to form an atlas whose surface emulates that of a pyramid. The decomposition of the surface into charts allows us to have *vision-dedicated* and *touch-dedicated* charts, which we fuse by deforming the vision charts around the touch charts.

An overview of our approach is highlighted in Figure 2. Touch signals are used to predict *touch charts* using a pre-trained fully convolutional network, while vision signals are used to define image features over the set of *touch and vision charts* using perceptual feature pooling [65]. This set of vision and touch charts are then iteratively deformed to obtain the 3D shape reconstruction.

## 3.1   Merging vision and touch through chart deformation and tactile consistency

We adapt the mesh deformation setup outlined in [65, 58] and employ a GCN to deform our set of vision and touch charts. The GCN learns a function $f_{\theta_1}^{chart}$ parameterized by $\theta_1 \subset \theta$ that predicts the *residual* position of the vertices within each chart $C_i$ through successive layers. Given a vertex $u$, each layer $l$ of the GCN updates the vertex's features $H_u^{l-1}$ as:

$$H_u^l = \sigma \left( W^l \left( \sum_{v \in \mathcal{N}_u \cup \{u\}} \frac{H_v^{l-1}}{\sqrt{|\mathcal{N}_u + 1||\mathcal{N}_v + 1|}} \right) + b^l \right) , \tag{1}$$

where $W^l$ and $b^l$ are the learnable weights and biases of the $l$-the layer, $\sigma$ is a non-linearity, and $\mathcal{N}_u$ are the neighbors of the vertex $u$. We initialize each vertex's features $H_u^0$ by concatenating vision features obtained by applying perceptual pooling to the input image, with the $(x, y, z)$ position of the vertex in space, and a binary feature indicating whether the vertex is within a successful touch chart. The function $f_{\theta_1}^{chart}$ is trained to minimize the Chamfer Distance [61] between two sets of points $S$ and $\hat{S}$ sampled from $O$ and $\{C_i\}_{i=1}^{n_c}$, respectively, over a dataset $\mathcal{D}$:

$$\sum_{i \in \mathcal{D}} \left( \sum_{p \in S^{(i)}} \min_{\hat{p} \in \hat{S}^{(i)}} \|p - \hat{p}\|_2^2 + \sum_{\hat{p} \in \hat{S}^{(i)}} \min_{p \in S^{(i)}} \|p - \hat{p}\|_2^2 \right) . \tag{2}$$

GCNs enforce the exchange of information between vertices, which belong to the same neighborhood at every layer to allow information to propagate throughout the graph (see Figure 9a). Vertices in independent charts, however, will never lie in the same neighborhood, and so no exchange of information between charts can occur. To allow for the exchange of information between *vision* charts, we initially arrange the vision charts $\{C_i^v\}_{i=1}^{n_{cv}}$ to form a closed sphere (with no chart overlap). Then, we update each vertex's neighborhood such that vertices on the boundaries of different charts are in each other's neighborhood if they initially touch (see Figure 9b). With this setup, the charts are able to effectively communicate throughout the deformation process, and move freely during the optimization to optimally emulate the target surface. This is advantageous over the standard mesh deformation scheme, which deforms an initial closed mesh, as the prediction is no longer constrained to any fixed surface genus. Moreover, to define the communication between *vision and touch* charts, and enable the touch charts to influence the position of the vision charts, a reference vertex from the center of each touch chart is elected to lie within the neighborhood of all boundary vertices of vision charts and vice versa (see Figure 9c). With this setup, every vision chart can communicate with other nearby vision charts, as well as the touch charts. This communication scheme allows local touch, and vision information to propagate and fuse over all charts.

The chart deformation occurs three times to refine the prediction [65, 58], with the predicted charts being fed back to the input of the GCN before producing a final prediction. In total, 95 vision charts are employed, each possessing 19 vertices and 24 faces each. Touch charts each posses 81 vertices and 128 faces. The GCN updates the positions of the charts, however the initial position of touch charts is enforced after every deformation step and in the final mesh, as their shape is known to be

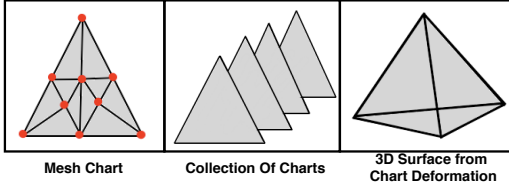

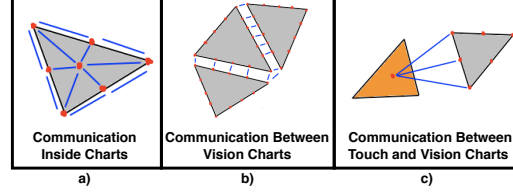

Figure 3: Structure of a chart along with how collections of charts can be deformed to produce a 3D surface, with vertices highlighted in red.

Figure 4: Communication within and between charts, with vertices highlighted in red, and communication between them highlighted in blue.

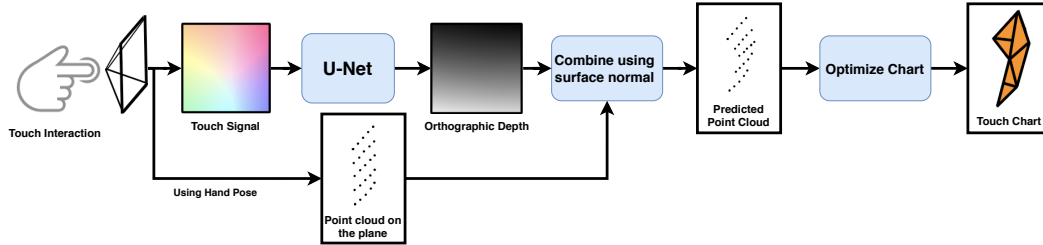

Figure 5: The pipeline for touch chart prediction from simulated touch readings. We start by predicting orthographic depth from touch reading image, then we combine orthographic depth with a point cloud sampled from the sensor plane using surface normal and obtain a predicted point cloud of local surface. To convert the predicted point cloud to touch chart by running an iterative optimization.

practically perfect. In this manner, the touch charts are fixed, and the vision charts learn to fill in the remaining surface around them through communication in the GCN. Touch charts corresponding to unsuccessful touches are initialized with simply the position of their finger tips at all vertices. This informs the model about where the fingers are in space, and so also where the object cannot be. Note that the position of unsuccessful touches is not enforced after their deformation.

### 3.2 Prediction of local touch charts

In this subsection, we describe how to obtain touch charts from touch signals produced using a gel-based sensor with high spatial resolution, such as the DIGIT [40]. To do this, we make note of what gel-based touch sensors truly observe: the impression of a surface through the gel. When untouched, the gel is lying perpendicular to the camera's perspective and at a fixed distance away. When touched by an object, that object's local surface is interpretable by the depth of the impression it makes across the plane of the gel. If we then want to recover this surface from the sensor, we simply need to interpret the touch signal in terms of the depth of the impression across this plane.

Figure 5 depicts the pipeline for local structure prediction. Using the finger position information $P$, we start by defining a grid of points $G_{init} \in \mathbb{R}^{100 \times 100 \times 3}$ of the same size and resolution as the sensor, lying on a perpendicular plane above it, which corresponds physically to the untouched gel of the sensor. Then, we apply a function $f_{\theta_2}^{touch}$, parameterized by $\theta_2 \subset \theta$, and represented as a fully convolutional network (U-Net-like model [55]) that takes as input the touch reading signal $R \in \mathbb{R}^{100 \times 100 \times 3}$ and predicts orthographic distance from each point to the surface [57]. This distance corresponds to the depth of the impression across the surface. Next, we transform this prediction into a point cloud $\hat{G}$ as:

$$\hat{G} = G_{init} + f_{\theta_2}^{touch}(R) * \hat{n}, \tag{3}$$

where $\hat{n}$ denotes the plane's unit normal. This transforms the grid of points to the shape of the gel across the impression, and so should match the local geometry which deformed it. To learn $\theta_2$, we minimize the Chamfer distance between the predicted point cloud $\hat{G}$ and the ground truth point cloud local to the touch site, $G$. After predicting the local point cloud $\hat{G}$, a local touch chart $C$ can be obtained by minimizing the Chamfer distance between points sampled from $C$ and $\hat{G}$.

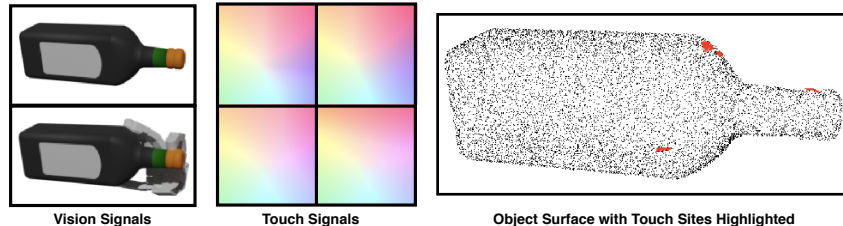

| Vision Signals | Touch Signals | Object Surface with Touch Sites Highlighted |

Figure 6: A data point from our dataset displaying an occluded (by hand) and unoccluded RGB image, 4 RGB images representing touch readings and a 3D object surface with touch sites highlighted.

## 4 A Visuotactile Dataset of Object-Grasp Interactions

To validate the model described in Section 3, we built a new dataset that aims to capture the interactions between a robotic hand and an object it is touching. We simulate these interactions using a Wonik's Allegro Hand [56] equipped with vision-based touch sensors [40] on each of its three fingers and thumb. We use objects from the 3D Warehouse [1], given its ubiquitous use in computer vision research, and to enable comparisons with previous vision-only work [58, 19, 21, 65].

An example instance of data collected from a single grasp is highlighted in Figure 6. We load example objects into the 3D robotics simulator Pybullet [11], place the hand randomly on its surface, and close its fingers attempting to produce contact between the sensors and some point on the object using inverse kinematics. To simulate the touch signal from each sensor, a grid of 10,000 points on the surface of the sensor are projected towards the object using sphere tracing [24] in Pytorch [49], defining a depth map from the sensor's perspective. We then define three lights (pure red, green and blue) around the boundary of the surface the depth map defines and a camera looking down at the surface from above. With this setup we then use the Phong reflection model [51] reflection model to compute the resulting simulated touch signal with resolution $100 \times 100 \times 3$. This process provides a quality approximation of how vision-based tactile sensors work and upon visual inspections the simulated images look plausible to a human expert. To acquire visual information from this interaction two images are rendered using Blender [10]: (1) a pre-interaction image of the object alone, and (2) an interaction image of an object occluded by the hand grasping it. Both images have resolution $256 \times 256 \times 3$ .Details with respect to the Allegro Hand, how the grasp simulations are performed in Pybullet, the rendering and scene settings in Blender, and the simulation of touch signals can be found in the supplemental materials.

The bottle, knife, cellphone, and rifle were chosen from the 3D Warehouse data due to their hand-held nature for a total of 1732 objects. From each grasp an occluded image, an unoccluded image, four simulated touch signals with a mask indicating if each touch was successful, the hand's current pose, a global point cloud of the object's shape, and four local point clouds defining each touch site are recorded. This information is visualized in Figure 6. From each object, five hand-object interactions, or grasps are recorded, and for each grasps at least one successful touch occurs, though on average 62.4% of touches are successful. We split the dataset into training and test sets with approximately a 90:10 ratio. Further details on the design and content of this dataset, together with in-depth statistics and analysis of the content, are provided in the supplemental materials.

## 5 Experimental Results

In the following section, we describe the experiments designed to validate our approach to 3D reconstruction that leverages both visual and haptic sensory information. We start by outlining our model selection process. Then, using our best model, we validate generalization of the complementary role of vision and touch for 3D shape reconstruction. We follow by examining the effect of increasing number of grasps and then measure the ability of our approach to effectively extrapolate around touch sites. For all experiments, details with respect to experiment design, optimization procedures, hardware used, runtime, and hyper-parameters considered can be found in the supplemental materials.

### 5.1 Complementarity of vision and touch: model selection and generalization

In the model selection, we compare our approach to three other modality fusion strategies on the validation set: (1) **Sphere-based**, where the chart-based initialization is replaced with a sphere-based

| Row | Model | Vision | Touch | Bottle | Knife | Cellphone | Rifle | Average |
|---|---|---|---|---|---|---|---|---|
| 1 | Sphere-based | U | ✓ | 0.775 | 0.572 | 1.262 | 0.643 | 0.813 |
| 2 | Chart-based (no copying) | U | ✓ | 0.741 | **0.538** | 1.141 | 0.603 | 0.756 |
| 3 | Chart-based (no comm.) | U | ✓ | **0.709** | 0.723 | 1.222 | 0.500 | 0.788 |
| 4 | Ours | U | ✓ | 0.741 | 0.676 | **1.116** | **0.473** | **0.751** |
| 5 | Sphere-based | O | ✓ | 0.985 | 0.692 | 1.270 | 1.023 | 0.992 |
| 6 | Chart-based (no copying) | O | ✓ | 0.953 | **0.656** | 1.176 | 0.892 | 0.919 |
| 7 | Chart-based (no comm.) | O | ✓ | 0.954 | 0.784 | 1.413 | 0.904 | 1.014 |
| 8 | Ours | O | ✓ | **0.872** | 0.685 | **1.142** | 0.806 | **0.876** |
| 9 | Sphere-based | U | ✗ | 0.816 | **0.561** | 1.322 | 0.667 | 0.841 |
| 10 | Ours | U | ✗ | **0.783** | 0.703 | **1.115** | **0.588** | **0.797** |
| 11 | Sphere-based | O | ✗ | 1.093 | **0.719** | 1.404 | 1.074 | 1.072 |
| 12 | Ours | O | ✗ | **0.994** | 0.831 | **1.301** | **0.956** | **1.020** |

Table 1: Model selection. We report the per-class Chamfer distance for the validation set together with average value. Note that O stands for *occluded* and U for *unoccluded*

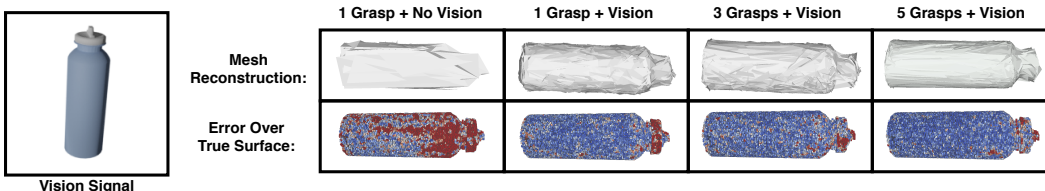

Figure 7: Reconstruction results of our method across different input modalities and number of grasps. For vision signal, we use an unoccluded RGB image.

one, and the sphere vertices contain a concatenation of projected vision features and touch features extracted from a simple CNN; (2) **Chart-based (no copying)**, where we remove the hard copying of local touch charts in the prediction; and (3) **Chart-based (no comm.)**, where we remove the communication between charts in the GCN and only copy them to the final prediction. For vision only inputs, we compare our model to the sphere-based model only. For all comparisons we consider both the occluded and unoccluded vision signals.

The results of model selection are presented in Table 1. We observe that: (1) the sphere-based model suffers from a decrease in average performance when compared to our model (see rows 1 vs 4, 5 vs 8, 9 vs 10, and 11, vs 12), (2) the copying of the local charts to the final prediction leads to performance boosts (see rows 2 vs 4, and 6 vs 8), and (3) the global prediction benefits from communication between touch and vision charts (see rows 3 vs 4, and 7 vs 8). Moreover, as expected, we notice a further decrease in average performance when comparing each unoccluded vision model with their occluded vision counterpart. Finally, for models leveraging vision and touch, we consistently observe an improvement w.r.t. their vision-only baselines, which particularly benefits our full chart-based approach. This improvement is especially noticeable when considering occluded vision, where touch information is able to enhance the reconstruction of sites occluded by the hand touching the object. To further validate our chart-based approach, its performance on single image 3D object reconstruction on [1] was evaluated and compared to an array of popular methods in this setting. The performance of our model here was highly competitive with that of state of the art methods and the results, details, and analysis of this experiment can be found in the supplemental materials.

In Table 2, we highlight the generalization of our best model by evaluating it on the test set for five 3D shape reconstruction set ups, namely occluded and unoccluded vision scenarios with and without touch. We notice that the improvement introduced by including the haptic modality generalizes to the test set, for both occluded and unoccluded vision signals. Moreover, we test our approach by removing the vision signal and optimizing it to recover the 3D shape using only tactile signals. In this case, we experience an increased global error of 3.050, compared to the next worse model with 1.074 error, demonstrating its difficulty to extrapolate without global context and further highlighting the locality of the touch signals. Finally, we display some example reconstructions that our best performing fusion models produce in Figure 7.

|  | Vision (occluded) | | Vision (unoccluded) | | |
|---|---|---|---|---|---|
| **Input** | Touch | No Touch | Touch | No touch | Touch only |
| **Ours** | 0.991 | 1.074 | 0.804 | 0.861 | 3.050 |

Table 2: Test set results for 3D reconstruction tasks with different input modalities: combination of touch readings and occluded or unoccluded vision signal.

| Class | Bottle | Knife | Cellphone | Rifle |
|---|---|---|---|---|
| C.D. | 0.0099 | 0.0136 | 0.0072 | 0.00749 |

Table 3: Chamfer distance per class for local point cloud prediction at each touch site.

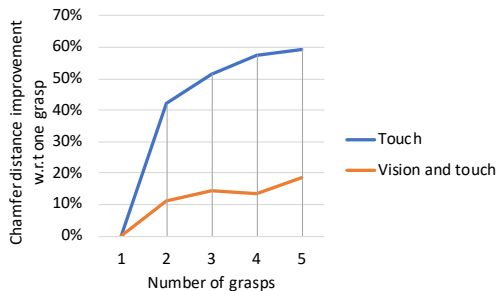

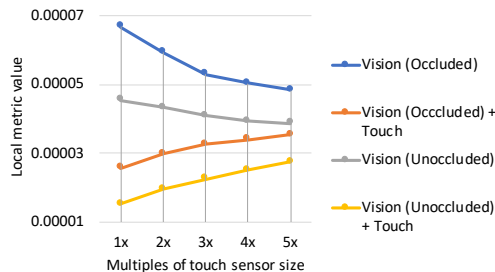

Figure 8: Multi-grasp experiment: we depict the Chamfer distance increase w.r.t. one grasp.

Figure 9: Local Chamfer distance at expanding distances around each touch site.

## 5.2 Going beyond a single grasp: Multi-grasp experiments

We design a further experiment in which we examine the effect of providing increasing number of grasps. The only practical change to the model here is that the number of touch charts increases by 4 for every additional grasp provided. This experiment is conducted using 1 to 5 grasps, and in both the touch-only setting where vision information is excluded and the unoccluded vision and touch setting. The results of this experiment are shown in Figure 8, where we demonstrate that by increasing the number of grasps provided, the reconstruction accuracy significantly improves both with and without the addition of unoccluded vision signals. Reconstruction results across different numbers of grasps can be viewed in Figure 7. From this experiment, it can be concluded that our model gains greater insight into the nature of an object by touching new areas on its surface.

## 5.3 From touch sensor readings to local structure prediction

Per-class reconstruction results at each touch site using the U-Net-based architecture are highlighted in Table 3. As expected, the reconstructions are practically perfect, when compared to the error incurred over full surfaces (smallest average global error of 0.804). The small errors incurred here are mainly due to the fact that predicted points are selected as belonging to the surface by observing differences in the touch signal and an untouched touch signal. This leads to overshooting and undershooting of the boundary of the touch, and consequently too large or too small predicted surfaces. A reconstruction result from this experiment is displayed in Figure 10.

Last, we design an experiment which examines how well the the target surface is reconstructed in expanding regions around each touch site. To do this, square rings of points of 1 to 5 times larger dimensions than the touch sensor are projected onto each object's surface at each touch site in order to produce increasingly distant regions around them. Then, the mean distance from these points to the closest point in the corresponding prediction is computed to determine how well these regions have been reconstructed. We perform this experiment with and without touch for both occluded and unoccluded vision models, and the results are shown in Figure 10. As expected, the vision-only models incur approximately the same loss at every plane size while models which leverage touch begin with a drastically lower loss and only slowly increase errors as the plane size increases. This experiment implies that the sharing of information between local and global charts allows for the propagation of touch information to regions around each touch site, suggesting a successful fusion of the complementary signals of vision and touch.

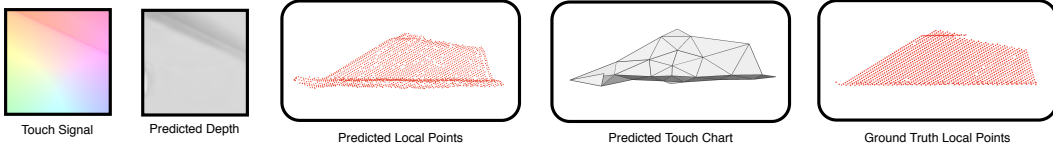

Figure 10: Local prediction of structure at a touch site, together with the touch chart.

## 5.4 Limitations

There exist a few limitations of this method which are worth addressing. The first noteworthy limitation is that in the final object predictions, charts sometimes poorly overlap creating a noisy boundary, as opposed to a more attractive smooth surface. This can be credited to the lack of natural regularizers for encouraging smoothness, which exist in more constrained methods [65, 58, 36]. This property can be observed in Figure 7. A second, and somewhat related limitation of the method is that charts are not forced to form a continuous connected surface. This occasionally leads to charts lying disconnected for the mainly body of the predicted surface. However, from our qualitative evaluation of 100 predicted objects in the test set, this only occurs 5% of the time, and has very little effect on the quality of the predictions. A final limitation is that our method requires full 3D scene information while training, and at test time requires full hand pose information. While trivial to acquire in our simulated environment, for application to real environments this data requirement is somewhat unrealistic.

## 6   Conclusion

In this paper, we explored the problem of 3D shape reconstruction from vision and touch. To do so, we introduced a dataset of simulated touch and vision signals, and proposed a chart-based approach that effectively exploits the complementary nature of both modalities, namely, the high fidelity local information from touch and the global information from vision. While some limitations to our approach exist such as potentially noisy or incomplete surface predictions, our results consistently highlight the benefit of combining both modalities to improve upon single modality baselines, and show the potential of using a chart-based approach to combine vision and touch signal in a principled way. The benefit of fusing vision and touch is further emphasized by the ability of our model to gracefully extrapolate around touch sites, and by the improved reconstruction accuracy when providing an increasing number of grasps, which suggests that the active sensing of visual and touch signals is a promising avenue to improve 3D shape reconstruction.

## Broader Impact

Our contributions allows for improved understanding of the three dimensional world in which we all live. The impact of this work lies mainly in the field of 3D object understanding, such as better 3D reconstruction of objects in simulated environments as well as potential improvements in shape understanding in real world robot-object manipulation. There are many benefits to using improved 3D object understanding, and it may prove especially useful for the fields of automation, robotic, computer graphics and augmented and virtual reality. Failures of these models could arise if automation tools are not properly introduced, and biases are not properly addressed. In particular, these models could result in poor recognition of 3D objects in diverse contexts, as has already been shown for 2D recognition systems. On the research side, to mitigate these risks, we encourage further investigation to outline the performance of 3D understanding systems in the wild in a diverse set of contexts and geographical locations, and to mitigate the associated performance drops.

## 7   Acknowledgments

We would like to acknowledge the NSERC Canadian Robotics Network, the Natural Sciences and Engineering Research Council, and the Fonds de recherche du Québec – Nature et Technologies for their funding support, as granted to the McGill University authors. We would also like to thank Scott Fujimoto and Shaoxiong Wang for their helpful feedback.

## Footnotes

*Correspondence to: ejsmith@fb.com and edward.smith@mail.mcgill.ca

[2]https://github.com/facebookresearch/3D-Vision-and-Touch

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
