[Supplementary Material]

# Supplementary Material

In the following sections, we provide additional details with respect to various elements of the paper which could not be fully expanded upon in the main paper. This begins with an in depth explanation of the proposed dataset, including its exact contents, and the manner in which they were produced. This is followed by a closer look into the various aspects of touch chart prediction including architectures, experimental procedures, hyper-parameters, and additional results. Finally, a comprehensive examination of the prediction of vision charts is provided, again with detailed explanations of architectures, experimental procedures, hyper-parameters, and additional results.

## 1 Visuotactile Dataset

This section describes the multi-modal dataset of simulated 3D touches, which this paper contributes and makes use of. This includes both the methods by which each component of the dataset was produced, and its exact contents.

### 1.1 Dataset content

For each object-grasp example in the dataset, the following are recorded:

- A dense point cloud of 10,000 points representing the object's surface, $S^{obj}$.

- Four local point clouds of at most 10,000 points, each representing the surface of the object at each touch site, $\{S_i^{loc}\}_{i=1}^4$.

- Four orthographic depth maps representing orthographic distance from the plane of the each touch sensor to any object geometry in front of them, $\{D_i\}_{i=1}^4$.

- Four simulated touch signals $T = [R_i, P_i, M_i]_{i=1}^{n_t=4}$, where $R_i$ is one touch sensor reading, $P_i$ its corresponding position and rotation in space, and $M_i$ a binary mask indicating whether the touch is successful (i.e. the sensor is in contact with the object).

- Two vision signals $V_u$ and $V_o$, corresponding to an image of the object alone (*unoccluded vision*) and an image of the object being grasped by the hand (*occluded vision*), respectively.

### 1.2 3D Objects and Hand

**3D Objects**: The 3D objects used for this dataset are from the 3D Warehouse data [1]. These are CAD objects and so possess geometry and texture information. For each object we want to grasp, $S^{obj}$ is extracted from its surface using the technique defined in [? ].

**Hand**: The Allegro hand [12] is used for grasping the objects. The URDF definition of this hand was altered to add the shape of a sensor to the finger tips and thumb tip of the hand. To make the hand easier to manipulate and render, its mesh components were altered by removing non-surface vertices and faces, and decimating the mesh. Note that this hand pre-processing has practically no impact on its behavior nor appearance.

### 1.3 Simulating Grasps

The 3D robotics simulator PyBullet [4] was used for producing the touch interactions. The process by which grasps are produced in PyBullet is displayed in Figure 1. First, the object and the Allegro hand are loaded into the simulator in a fixed pose (see first two images from left to right). The Allegro hand is then placed such that its palm is tangent to a random point on the object's surface (see third image). The distance from the object and the rotation of the hand on the tangent plane is also random. Using inverse kinematics, the joints of the hand are then rotated such that each of the hand's touch

| Load Hand into Scene | Load Object into Scene | Move Hand to Surface | Grasp the Object |

Figure 1: Visualization of the procedure used to create grasps in PyBullet.

sensors meet the surface of the object, so long as physical integration of the hand and object allow it (see fourth image). This is repeated multiple times until sufficiently many grasps are recorded with successful touches. The pose information $P_i$, and the masks $M_i$ from the best 5 grasps (with respect to the number of successful touches) are saved for use in the dataset.

### 1.4 Simulating Vision

To simulate the vision signals for each grasp, two RGB images $V_u$ and $V_o$ are rendered using Blender [3]. The object is placed at position [0, 0, 0.6] in its canonical 3D Warehouse pose, and the camera is placed at position [0.972, 0.461, 0.974], with rotation [70.458°, 4.940°, 113.540°]. Both images have resolution of $256 \times 256 \times 3$, and are produced with constant lighting from a single lamp.

### 1.5 Simulating Touch

For each grasp, to simulate the touch signal, $T$, at a successful touch site, and in particular its readings $R_i$, the 3D mesh of its corresponding object is first loaded into PyTorch [9]. Then, a $100 \times 100$ grid of points is created with the same size, position, shape, and orientation as the sensor. The dimensions of this grid define the resolution of the sensor: $100 \times 100$ pixels. The grid of points is projected orthogonally towards the surface of the object using sphere tracing, and so halts exactly at the touch site of the sensor. The distance these points move during this projection defines an orthographic depth map, $D_i$, for each touch sensor. The final position of the points defines the local structure of the surface, which the sensor interacts with. The depth maps of each touch site are saved, along with the points in the point cloud which correspond to depths smaller than the true depth of the sensor. We find the depth of the impression into the touch sensor as:

$$D_i' = ReLU(w - D_i), \tag{1}$$

where $w$ is the depth of the sensor. Then, each position $x, y$ in $D_i'$ is projected into a 3D point cloud $S_i^{loc}$ as follows: $[x/100, y/100, D_i'[x, y]]$.

To obtain the simulated RGB touch reading $R_i$ from $S_i^{loc}$, three lights of pure red, green and blue are defined in a triangular shape above this surface at positions $P_r$, $P_g$, and $P_b$. We then use the Phong reflection model [10], where we assume zero specular or ambient lighting. The intensity values for the red colour channel of the simulated touch reading, $R_i \in \mathbb{R}^{100 \times 100 \times 3}$, are then defined as:

$$R_i = \lambda * \hat{n} * \hat{l}, \tag{2}$$

where $\lambda$ is the diffuse reflection constant, $\hat{n}$ is the unit normal of the plane (broadcasted for shape compatibility), defined as

$$n = \left[ -\frac{\mathrm{d}D_i'}{\mathrm{d}x}, -\frac{\mathrm{d}D_i'}{\mathrm{d}y}, 1 \right], \tag{3}$$

$$\hat{n} = \frac{n}{\|n\|}, \tag{4}$$

and $\hat{l}$ is the normalized light direction, defined as

$$\hat{l} = \frac{P_r - S_i^{loc}}{\|P_r - S_i^{loc}\|}, \tag{5}$$

where $P_r$ is broadcast for shape compatibility. The intensity values for the green and blue colour channels are defined in the same manner.

| Class | Objects | Grasps | Touches | % Successful |
|---|---|---|---|---|
| Bottle | 487 | 2435 | 9740 | 71.8 |
| Knife | 416 | 2080 | 8320 | 54.1 |
| Cellphone | 493 | 2465 | 9860 | 67.2 |
| Rifle | 335 | 1675 | 6700 | 53.6 |

Table 1: Per-Class dataset statistics of the number of objects, grasps, touches and percentage of successful touches in each class.

## 1.6 Dataset Statistics

Five classes from 3D Warehouse were used in the dataset: the bottle, knife, cellphone, and rifle classes, for a total of 1731 objects. We split the dataset into a training set with 1298 objects, a validation set with 258 objects, and a test set with 175 objects. Statistics on the size, number of grasps, number of touches, and the percentage of those that were successful are provided in in Table 1. With respect to the distribution of successful touches over grasps, 9.47% of grasps possess only 1 successful touch, 26.08% possess 2, 53.83% possess 3, and finally 10.61% possess 4. Additional dataset examples are displayed in Figure 2.

## 2 Local Touch Chart Predictions

In the following section, additional details are provided with respect to how predictions of local touch charts are created. These include details surrounding the architecture of models used, the range of hyperparameters considered, optimization details, additional results, runtime, and hardware used.

### 2.1 Model Architecture Details

To predict local charts we first predict a depth map. As described in the paper, a U-Net-based architecture [11] is leveraged for this task. The exact architecture for this network is displayed in Table 5.

### 2.2 Optimization Details

The model was trained using the Adam optimizer [8] with learning rate 5e-5 on 8 Tesla V100 GPUs with 16 CPU cores each, and batch size of 32. The learning rate for this experiment was tuned on the following grid [0.001, 0.0001, 0.00005, 0.00003, 0.00001]. The model was trained for a maximum number of 114 epochs (a total of 3 hours), it was evaluated every epoch on the validation set, and the best performing model across these evaluations was selected.

Moreover, it is worth noting that when optimizing the model's parameters, we compute the loss only for those positions in the touch sensor that interacted with the object. To do that, we first calculate the difference between the touch reading (sensor-object interaction) and an untouched sensor reading (no sensor-object interaction), and then compute the pixelwise $\ell_2^2$ of the resulting differences. Finally, we apply a threshold of 0.001 and only consider those positions with a greater value.

### 2.3 Converting Point Clouds to Charts

As explained in the paper, the trained U-Net-based model produces a local point cloud for each touch signal in the dataset. Each point cloud is then used to produce a touch chart. To do this, a planar, triangular mesh with 81 vertices and 128 faces is first placed in the same position, orientation, shape, and size as the touch sensor which produced the touch signal. Then, using Adam[8] and learning rate 0.003, the position of the vertices in the chart is optimized so that it emulates the surface of the point cloud. The loss used for this optimization is the Chamfer distance between the point cloud and points uniformly sampled on the surface of the chart as in [13]. The optimization scheme halts when the loss is lower than 0.0006.

### 2.4 Additional Results

Additional visual reconstruction results are displayed in Figure 3. From these visualization, it can be seen that the predicted point clouds almost perfectly match the ground truth point clouds, though with a small degree of extrapolation beyond the observed surface. It can also be observed that the corresponding chart predictions almost perfectly match the predicted point clouds.

| Vision Signals | Touch Signals | Object Surface with Touch Sites Highlighted |
|---|---|---|

| Vision Signals | Touch Signals | Object Surface with Touch Sites Highlighted |
|---|---|---|

| Vision Signals | Touch Signals | Object Surface with Touch Sites Highlighted |
|---|---|---|

| Vision Signals | Touch Signals | Object Surface with Touch Sites Highlighted |
|---|---|---|

Figure 2: Visualization examples from the dataset showing an occluded (by hand) and unoccluded RGB image, 4 RGB images representing touch readings and a 3D object surface with touch sites highlighted.

| Touch Signal | Predicted Depth | Predicted Local Points | Predicted Touch Chart | Ground Truth Local Points |
|---|---|---|---|---|

Figure 3: Local predictions of structure at each touch chart together with the corresponding charts produced from them.

# 3 Global Vision Chart Predictions

In the following section, additional details are provided with respect to how vision charts are deformed around known touch charts, such that their combination emulates the target surface. These include details such as the models' architectures, the range of hyperparameters considered, optimization details, additional results, runtime, and hardware used.

## 3.1 Chart Feature Initialization

Vision and touch charts must have features defined over their vertices before they can be combined and passed to the Graph Convolutional Network (GCN) to be deformed. Three types of features are defined over all charts: image features, position of the vertex, and a masking feature indicating if the chart corresponds to a successful touch or not. For touch charts, the position and mask feature of each of their vertices are predefined. The initial position of vision charts is defined such that they combine to form a closed sphere, only touching at their boundary. This arrangement is highlighted in Figure 4. Their mask feature is set to 0 as they do not correspond to successful touches. The image features of both vision and touch charts are defined using perceptual feature pooling [15]. Here images are passed through a Convolutional Neural Network (CNN) and feature maps from intermediate layers are extracted. For any given vertex, its 3D position in space is projected onto the 2D plane of the input image using known camera parameters. The location of this projection in the pixel space of the image corresponds exactly to a position in each feature map. The vertex's image features are then defined as the bilinear interpolation between the four closest features to the projection in each feature map.

## 3.2 Model Architecture Details

Two networks are used to deform the positions of vision charts. The first is the CNN which defines image features for perceptual feature pooling, and the second is the GCN which updates the vertex positions. The network architectures for each model evaluated on the test set are displayed in Tables 7, 8, 9, 10, and 6. The GCN layers in each architecture are zero-neighbor layers as defined in [13].

## 3.3 Optimization Details

Each model type was trained using Adam [8] with learning rate 3e-5 and batch size 16 on a Tesla V100 GPU with 16 CPU cores. Each model was allowed to train for a maximum of 260 epoch (roughly 12 hours), was evaluated every epoch on the validation set and the best performing model across these evaluations was selected. Models were early-stopped if they failed to improve on the

Figure 4: Initial positions of vision charts in a closed sphere. Charts have been separated slightly to improve their distinction.

| Multiples of Touch Sensor Size | x1 | x2 | x3 | x4 | x5 |
|---|---|---|---|---|---|
| Occluded Vision | 6.687e-5 | 5.938e-5 | 5.305e-5 | 5.038e-5 | 4.841e-5 |
| Occluded Vision + Touch | 2.569e-5 | 2.970e-5 | 3.257e-5 | 3.393e-5 | 3.528e-5 |
| Unoccluded Vision | 4.553e-5 | 4.333e-5 | 4.080e-5 | 3.922e-5 | 3.866e-5 |
| Unoccluded Vision + Touch | 1.510e-5 | 1.944e-5 | 2.243e-5 | 2.494e-5 | 2.736e-5 |

Table 2: Local Chamfer distance in increasingly large square rings around each touch sites.

validation set for 70 epochs. The best performing model with occluded vision and touch was selected on epoch 113. The best performing model with only occluded vision was selected on epoch 114. The best performing model with unoccluded vision and touch was selected on epoch 122. The best performing model with only unoccluded vision was selected on epoch 99. The best performing model with only touch was selected on epoch 90.

## 3.4 Hyperparameter Details

The hyper-parameters tuned for the experiments in this setting were the learning rate, the number of layers in the CNN, the number of layers in the GCN, and the number of features per GCN layer. The possible settings for the learning rate were [1e-4, 3e-5, 1e-5]. The possible settings for the number of CNN layers were [12, 15, 18]. The possible settings for the number of GCN layers were [15, 20, 25]. The possible settings for the number of features per GCN layer were [150, 200, 250].

## 3.5 Additional Results

Additional reconstruction results for each class are visualized in Figure 5. Numerical results for the mutli-grasp experiment are displayed in Table 3. Numerical results for the experiment which examined the local Chamfer distance at expanding distances around each touch site are displayed in Table 2.

## 3.6 Single Image 3D Object Reconstruction

As mentioned in the main paper, and as a sanity check, the chart-based approach to 3D object reconstruction was also applied to the task of single image 3D object reconstruction to validate that it is competitive with other vision exclusive methods for 3D shape reconstruction. We used the 3D models with rendered images from [2], and compared using the evaluation setup released by [6]. The model was trained for a maximum of 40 epochs (roughly 3 days of training), was evaluated after each

| Number of Grasps | 1 | 2 | 3 | 4 | 5 |
|---|---|---|---|---|---|
| Unoccluded Vision + Touch | 0.804 | 0.714 | 0.689 | 0.695 | 0.654 |
| Touch | 3.05 | 1.769 | 1.479 | 1.296 | 1.237 |

Table 3: Chamfer distance when increasing the number of grasps provided to the models.

| Vision Signal | 1 Grasp + No Vision | 1 Grasp + Vision | 3 Grasps + Vision | 5 Grasps + Vision |

Figure 5: Reconstruction results of our method for each class across different input modalities and number of grasps. For vision signal, we use an unoccluded RGB image.

epoch, and the best performing model across these evaluations was selected. The model was trained with the Adam optimizer [8] with learning rate e-5 and batch size 64 on a Tesla V100 GPU with 16 CPU cores. In this set up, we removed touch charts from our prediction pipeline and used exclusively vision signals. The architecture used for this experiment is displayed in Table 11.

We highlight the results of the evaluation in Table 4. The Chamfer Distance shown is the same metric as in the main paper, however, the scaling and density of points is different and so not comparable to other experiments. For a given distance threshold $\tau$, $F1^{\tau}$ is the harmonic mean of the precision (percentage of predicted points with distance at most $\tau$ from any ground truth point) and recall (percentage of ground truth points with distance at most $\tau$ from any predicted point) of predicted and ground truth point clouds. The table demonstrates that we are competitive with other vision based approaches to 3D shape reconstruction, only failing to outperform the newly released MeshRCNN algorithm [6]. It should be noted that our approach has not been heavily tuned for this specific dataset or task, and so failing to overtake the most recent state of the art method is not wholly surprising.

Table 4: Single image 3D shape reconstructing results on the 3D Warehouse Dataset. This evaluation is performed using the evaluation standard from [6] and [15].

| | Chamfer Distance($\downarrow$) | $F1^{\tau}$ ($\uparrow$) | $F1^{2\tau}$ ($\uparrow$) |
|---|---|---|---|
| N3MR [7] | 2.629 | 3.80 | 47.72 |
| 3D-R2N2 [2] | 1.445 | 39.01 | 54.62 |
| PSG [5] | 0.593 | 48.58 | 69.78 |
| MVD [**?** ] | - | 66.39 | - |
| GEOMetrics [13] | - | 67.37 | - |
| Pixel2Mesh [15] | 0.463 | 67.89 | 79.88 |
| MeshRCNN [6] (Pretty) | 0.391 | 69.83 | 81.76 |
| MeshRCNN [6] (Best) | 0.306 | 74.84 | 85.75 |
| Ours | 0.369 | 69.52 | 82.33 |

Table 5: Architecture for the U-Net style network used to predict point cloud positions for our local touch charts.

| Index | Input | Operation | Output Shape |
|---|---|---|---|
| (1) | Input | Conv ($3 \times 3$) + BN + ReLU | $64 \times 100 \times 100$ |
| (2) | (1) | Conv ($3 \times 3$) + BN + ReLU | $64 \times 100 \times 100$ |
| (3) | (2) | MaxPooling ($2 \times 2$) | $64 \times 50 \times 50$ |
| (4) | (3) | Conv ($3 \times 3$) + BN + ReLU | $128 \times 50 \times 50$ |
| (5) | (4) | Conv ($3 \times 3$) + BN + ReLU | $128 \times 50 \times 50$ |
| (6) | (5) | MaxPooling ($2 \times 2$) | $128 \times 25 \times 25$ |
| (7) | (6) | Conv ($3 \times 3$) + BN + ReLU | $256 \times 25 \times 25$ |
| (8) | (7) | Conv ($3 \times 3$) + BN + ReLU | $256 \times 25 \times 25$ |
| (9) | (8) | MaxPooling ($2 \times 2$) | $256 \times 12 \times 12$ |
| (10) | (8) | Conv ($3 \times 3$) + BN + ReLU | $512 \times 12 \times 12$ |
| (11) | (10) | Conv ($3 \times 3$) + BN + ReLU | $512 \times 12 \times 12$ |
| (12) | (11) | MaxPooling ($2 \times 2$) | $512 \times 6 \times 6$ |
| (13) | (12) | Conv ($3 \times 3$) + BN + ReLU | $1024 \times 6 \times 6$ |
| (14) | (13) | Conv ($3 \times 3$) + BN + ReLU | $1024 \times 6 \times 6$ |
| (15) | (14) | DeConv ($2 \times 2$) | $512 \times 12 \times 12$ |
| (16) | (15) (11) | Concatenate | $1024 \times 12 \times 12$ |
| (17) | (16) | Conv ($2 \times 2$) + BN + ReLU | $512 \times 12 \times 12$ |
| (18) | (17) | DeConv ($2 \times 2$) | $256 \times 25 \times 25$ |
| (19) | (18) (8) | Concatenate | $512 \times 25 \times 25$ |
| (20) | (19) | Conv ($2 \times 2$) + BN + ReLU | $256 \times 25 \times 25$ |
| (21) | (20) | DeConv ($2 \times 2$) | $128 \times 50 \times 50$ |
| (22) | (21) (5) | Concatenate | $256 \times 50 \times 50$ |
| (23) | (22) | Conv ($2 \times 2$) + BN + ReLU | $128 \times 50 \times 50$ |
| (24) | (23) | DeConv ($2 \times 2$) | $64 \times 100 \times 100$ |
| (25) | (24) (2) | Concatenate | $128 \times 100 \times 100$ |
| (26) | (25) | Conv ($2 \times 2$) + BN + ReLU | $64 \times 100 \times 100$ |
| (27) | (26) | Conv ($1 \times 1$) | $1 \times 100 \times 100$ |

Table 6: Architecture for deforming charts with touch information only ($|V|$x3).

| Index | Input | Operation | Output Shape |
|---|---|---|---|
| (1) | Vertex Inputs | GCN Layer | $|V| \times 250$ |
| (2) | (1) | GCN Layer | $|V| \times 250$ |
| ... | ... | ... | ... |
| (15) | (14) | GCN Layer | $|V| \times 3$ |

Table 7: Architecture for deforming charts with occluded vision signals and touch information. The input to this model is an RGB image (4x256x256), and vertex features (|V|x4). BN refers to batch normalization.

| Index | Input | Operation | Output Shape |
|---|---|---|---|
| (1) | Image Input | Conv $(3 \times 3)$ + BN + ReLU | $16 \times 127 \times 127$ |
| (2) | (1) | Conv $(3 \times 3)$ + BN + ReLU | $16 \times 125 \times 125$ |
| (3) | (2) | Conv $(3 \times 3)$ + BN + ReLU | $16 \times 123 \times 123$ |
| (4) | (3) | Conv $(3 \times 3)$ + BN + ReLU | $16 \times 121 \times 121$ |
| (5) | (4) | Conv $(3 \times 3)$ + BN + ReLU | $16 \times 119 \times 119$ |
| (6) | (5) | Conv $(3 \times 3)$ (stride 2) + BN + ReLU | $32 \times 59 \times 59$ |
| (7) | (6) | Conv $(3 \times 3)$ + BN + ReLU | $32 \times 57 \times 57$ |
| (8) | (7) | Conv $(3 \times 3)$ + BN + ReLU | $32 \times 55 \times 55$ |
| (9) | (8) | Conv $(3 \times 3)$ + BN + ReLU | $32 \times 53 \times 53$ |
| (10) | (9) | Conv $(3 \times 3)$ + BN + ReLU | $32 \times 51 \times 51$ |
| (11) | (10) | Conv $(3 \times 3)$ (stride 2) + BN + ReLU | $64 \times 25 \times 25$ |
| (12) | (11) | Conv $(3 \times 3)$ + BN + ReLU | $64 \times 23 \times 23$ |
| (13) | (12) | Conv $(3 \times 3)$ + BN + ReLU | $64 \times 21 \times 21$ |
| (14) | (13) | Conv $(3 \times 3)$ + BN + ReLU | $64 \times 19 \times 19$ |
| (15) | (14) | Conv $(3 \times 3)$ + BN + ReLU | $64 \times 17 \times 17$ |
| (16) | (15) | Conv $(3 \times 3)$ (stride 2) + BN + ReLU | $128 \times 8 \times 8$ |
| (17) | (16) | Conv $(3 \times 3)$ + BN + ReLU | $128 \times 6 \times 6$ |
| (18) | (17) | Conv $(3 \times 3)$ + BN + ReLU | $128 \times 4 \times 4$ |
| (19) | (10) (15) (18) | perceptual feature pooling | $|V| \times 224$ |
| (20) | (19) Vertex Input | Concatenate | $|V| \times 228$ |
| (21) | (20) | GCN Layer | $|V| \times 250$ |
| (22) | (21) | GCN Layer | $|V| \times 250$ |
| ... | ... | ... | ... |
| (41) | (40) | GCN Layer | $|V| \times 3$ |

Table 8: Architecture for deforming charts with occluded vision signals without touch information. The input to this model is an RGB image (4x256x256), and vertex features (|V|x3). BN refers to batch normalization.

| Index | Input | Operation | Output Shape |
|---|---|---|---|
| (1) | Image Input | Conv $(3 \times 3)$ + BN + ReLU | $16 \times 127 \times 127$ |
| (2) | (1) | Conv $(3 \times 3)$ + BN + ReLU | $16 \times 125 \times 125$ |
| (3) | (2) | Conv $(3 \times 3)$ + BN + ReLU | $16 \times 123 \times 123$ |
| (4) | (3) | Conv $(3 \times 3)$ + BN + ReLU | $16 \times 121 \times 121$ |
| (5) | (4) | Conv $(3 \times 3)$ + BN + ReLU | $16 \times 119 \times 119$ |
| (6) | (5) | Conv $(3 \times 3)$ (stride 2) + BN + ReLU | $32 \times 59 \times 59$ |
| (7) | (6) | Conv $(3 \times 3)$ + BN + ReLU | $32 \times 57 \times 57$ |
| (8) | (7) | Conv $(3 \times 3)$ + BN + ReLU | $32 \times 55 \times 55$ |
| (9) | (8) | Conv $(3 \times 3)$ + BN + ReLU | $32 \times 53 \times 53$ |
| (10) | (9) | Conv $(3 \times 3)$ + BN + ReLU | $32 \times 51 \times 51$ |
| (11) | (10) | Conv $(3 \times 3)$ (stride 2) + BN + ReLU | $64 \times 25 \times 25$ |
| (12) | (11) | Conv $(3 \times 3)$ + BN + ReLU | $64 \times 23 \times 23$ |
| (13) | (12) | Conv $(3 \times 3)$ + BN + ReLU | $64 \times 21 \times 21$ |
| (14) | (13) | Conv $(3 \times 3)$ + BN + ReLU | $64 \times 19 \times 19$ |
| (15) | (14) | Conv $(3 \times 3)$ + BN + ReLU | $64 \times 17 \times 17$ |
| (16) | (15) | Conv $(3 \times 3)$ (stride 2) + BN + ReLU | $128 \times 8 \times 8$ |
| (17) | (16) | Conv $(3 \times 3)$ + BN + ReLU | $128 \times 6 \times 6$ |
| (18) | (17) | Conv $(3 \times 3)$ + BN + ReLU | $128 \times 4 \times 4$ |
| (19) | (10) (15) (18) | perceptual feature pooling | $|V| \times 224$ |
| (20) | (19) Vertex Input | Concatenate | $|V| \times 227$ |
| (21) | (20) | GCN Layer | $|V| \times 200$ |
| (22) | (21) | GCN Layer | $|V| \times 200$ |
| ... | ... | ... | ... |
| (41) | (40) | GCN Layer | $|V| \times 3$ |

Table 9: Architecture for deforming charts with unoccluded vision signals and touch information. The input to this model is an RGB image (4x256x256), and vertex features (|V|x4). BN refers to batch normalization.

| Index | Input | Operation | Output Shape |
|---|---|---|---|
| (1) | Image Input | Conv ($3 \times 3$) + BN + ReLU | $16 \times 127 \times 127$ |
| (2) | (1) | Conv ($3 \times 3$) + BN + ReLU | $16 \times 125 \times 125$ |
| (3) | (2) | Conv ($3 \times 3$) + BN + ReLU | $16 \times 123 \times 123$ |
| (4) | (3) | Conv ($3 \times 3$) + BN + ReLU | $16 \times 121 \times 121$ |
| (5) | (4) | Conv ($3 \times 3$) + BN + ReLU | $16 \times 119 \times 119$ |
| (6) | (5) | Conv ($3 \times 3$) (stride 2) + BN + ReLU | $32 \times 59 \times 59$ |
| (7) | (6) | Conv ($3 \times 3$) + BN + ReLU | $32 \times 57 \times 57$ |
| (8) | (7) | Conv ($3 \times 3$) + BN + ReLU | $32 \times 55 \times 55$ |
| (9) | (8) | Conv ($3 \times 3$) + BN + ReLU | $32 \times 53 \times 53$ |
| (10) | (9) | Conv ($3 \times 3$) + BN + ReLU | $32 \times 51 \times 51$ |
| (11) | (10) | Conv ($3 \times 3$) (stride 2) + BN + ReLU | $64 \times 25 \times 25$ |
| (12) | (11) | Conv ($3 \times 3$) + BN + ReLU | $64 \times 23 \times 23$ |
| (13) | (12) | Conv ($3 \times 3$) + BN + ReLU | $64 \times 21 \times 21$ |
| (14) | (13) | Conv ($3 \times 3$) + BN + ReLU | $64 \times 19 \times 19$ |
| (15) | (14) | Conv ($3 \times 3$) + BN + ReLU | $64 \times 17 \times 17$ |
| (16) | (15) | Conv ($3 \times 3$) (stride 2) + BN + ReLU | $128 \times 8 \times 8$ |
| (17) | (16) | Conv ($3 \times 3$) + BN + ReLU | $128 \times 6 \times 6$ |
| (18) | (17) | Conv ($3 \times 3$) + BN + ReLU | $128 \times 4 \times 4$ |
| (19) | (10) (15) (18) | perceptual feature pooling | $|V| \times 224$ |
| (20) | (19) Vertex Input | Concatenate | $|V| \times 228$ |
| (21) | (20) | GCN Layer | $|V| \times 200$ |
| (22) | (21) | GCN Layer | $|V| \times 200$ |
| ... | ... | ... | ... |
| (46) | (45) | GCN Layer | $|V| \times 3$ |

Table 10: Architecture for deforming charts with unoccluded vision signals without touch information. The input to this model is an RGB image (4x256x256), and vertex features (|V|x3). BN refers to batch normalization.

| Index | Input | Operation | Output Shape |
|---|---|---|---|
| (1) | Image Input | Conv ($3 \times 3$) + BN + ReLU | $16 \times 127 \times 127$ |
| (2) | (1) | Conv ($3 \times 3$) + BN + ReLU | $16 \times 125 \times 125$ |
| (3) | (2) | Conv ($3 \times 3$) + BN + ReLU | $16 \times 123 \times 123$ |
| (4) | (3) | Conv ($3 \times 3$) (stride 2) + BN + ReLU | $32 \times 61 \times 61$ |
| (5) | (4) | Conv ($3 \times 3$) + BN + ReLU | $32 \times 59 \times 59$ |
| (6) | (5) | Conv ($3 \times 3$) + BN + ReLU | $32 \times 57 \times 57$ |
| (7) | (6) | Conv ($3 \times 3$) (stride 2) + BN + ReLU | $64 \times 28 \times 28$ |
| (8) | (7) | Conv ($3 \times 3$) + BN + ReLU | $64 \times 26 \times 26$ |
| (9) | (8) | Conv ($3 \times 3$) + BN + ReLU | $64 \times 24 \times 24$ |
| (10) | (9) | Conv ($3 \times 3$) (stride 2) + BN + ReLU | $128 \times 11 \times 11$ |
| (11) | (10) | Conv ($3 \times 3$) + BN + ReLU | $128 \times 9 \times 9$ |
| (12) | (11) | Conv ($3 \times 3$) + BN + ReLU | $128 \times 7 \times 7$ |
| (13) | (3) (6) (9) (11) | perceptual feature pooling | $|V| \times 240$ |
| (14) | (13) Vertex Input | Concatenate | $|V| \times 243$ |
| (15) | (14) | GCN Layer | $|V| \times 250$ |
| (16) | (15) | GCN Layer | $|V| \times 250$ |
| ... | ... | ... | ... |
| (35) | (34) | GCN Layer | $|V| \times 3$ |

Table 11: Architecture for chart deformation in the single image 3D object reconstruction experiment on the 3D Warehouse. The input to this model is an RGB image (3x256x256), and vertex features (|V|x3). IN refers to instance normalization [14].

| Index | Input | Operation | Output Shape |
|-------|-------|-----------|--------------|
| (1) | Image Input | Conv $(3 \times 3)$ + IN + ReLU | $16 \times 69 \times 69$ |
| (2) | (1) | Conv $(3 \times 3)$ + IN + ReLU | $16 \times 69 \times 69$ |
| (3) | (2) | Conv $(3 \times 3)$ + IN + ReLU | $16 \times 69 \times 69$ |
| (4) | (3) | Conv $(3 \times 3)$ + IN + ReLU | $16 \times 69 \times 69$ |
| (5) | (4) | Conv $(3 \times 3)$ + IN + ReLU | $16 \times 69 \times 69$ |
| (6) | (5) | Conv $(3 \times 3)$ (stride 2) + IN + ReLU | $32 \times 35 \times 35$ |
| (7) | (6) | Conv $(3 \times 3)$ + IN + ReLU | $32 \times 35 \times 35$ |
| (8) | (7) | Conv $(3 \times 3)$ + IN + ReLU | $32 \times 35 \times 35$ |
| (9) | (8) | Conv $(3 \times 3)$ + IN + ReLU | $32 \times 35 \times 35$ |
| (10) | (9) | Conv $(3 \times 3)$ + IN + ReLU | $32 \times 35 \times 35$ |
| (11) | (10) | Conv $(3 \times 3)$ (stride 2) + IN + ReLU | $64 \times 18 \times 18$ |
| (12) | (11) | Conv $(3 \times 3)$ + IN + ReLU | $64 \times 18 \times 18$ |
| (13) | (12) | Conv $(3 \times 3)$ + IN + ReLU | $64 \times 18 \times 18$ |
| (14) | (13) | Conv $(3 \times 3)$ + IN + ReLU | $64 \times 18 \times 18$ |
| (15) | (14) | Conv $(3 \times 3)$ + IN + ReLU | $64 \times 18 \times 18$ |
| (16) | (15) | Conv $(3 \times 3)$ (stride 2) + IN + ReLU | $128 \times 9 \times 9$ |
| (17) | (16) | Conv $(3 \times 3)$ + IN + ReLU | $128 \times 9 \times 9$ |
| (18) | (17) | Conv $(3 \times 3)$ + IN + ReLU | $128 \times 9 \times 9$ |
| (19) | (18) | Conv $(3 \times 3)$ + IN + ReLU | $128 \times 9 \times 9$ |
| (20) | (19) | Conv $(3 \times 3)$ + IN + ReLU | $128 \times 9 \times 9$ |
| (21) | (20) | Conv $(3 \times 3)$ (stride 2) + IN + ReLU | $256 \times 5 \times 5$ |
| (22) | (21) | Conv $(3 \times 3)$ + IN + ReLU | $256 \times 5 \times 5$ |
| (23) | (22) | Conv $(3 \times 3)$ + IN + ReLU | $256 \times 5 \times 5$ |
| (24) | (23) | Conv $(3 \times 3)$ + IN + ReLU | $256 \times 5 \times 5$ |
| (25) | (24) | Conv $(3 \times 3)$ + IN + ReLU | $256 \times 5 \times 5$ |
| (26) | (25) (22) (19) | perceptual feature pooling | $|V| \times 896$ |
| (27) | (26) | Linear + ReLU | $|V| \times 250$ |
| (28) | (27) Vertex Input | Concatenate | $|V| \times 253$ |
| (29) | (28) | GCN Layer | $|V| \times 250$ |
| (30) | (29) | GCN Layer | $|V| \times 250$ |
| ... | ... | ... | ... |
| (53) | (52) | GCN Layer | $|V| \times 3$ |