[Reviews · NeurIPS 2020]

Review 1

Summary and Contributions: This paper presents a method to reconstruct objects from a single image with additional input from a GelSight-type simulated sensor. The key insight of the paper is that images provide global context about shape while touch provides more local information. The paper achieves this using an extension of AtlasNet to both vision and touch sensing outputs. Experiments show that a combination of vision and touch works best.

Strengths: The paper has several strengths: - It solves an important problem of broad relevance to robotics and vision. - The idea of using vision for global context and touch for local details is nice (although there is no evidence that this happens in humans as the paper claims, see below) - The idea of using multiple charts together with graph networks to get more accurate shape is interesting. - The paper is well motivated and written.

Weaknesses: Below, I provide a list of limitations of the work, questions, and comments. - While I appreciated the idea of using vision for global context and touch for local details, it is a stretch to claim that this is what happens in humans as is claimed in the abstract and intro. I would suggest removing this claim -- in machine learning we seem to anthropomorphize our algorithms with little evidence. "...touch provides high fidelity localized information while vision provides complementary global context" A counterexample to this claim is the case of congenitally blind people who seem to have no problem describing global context of things they touch. See "Imagery in the congenitally blind: How visual are visual images?", Zimler and Keenan 1983 - The way the paper presents the idea of using charts makes it seem like it is a novel contribution, but in reality it is built on top of AtlasNet, who also use the term chart to describe their method. In fact, a follow up paper to AtlasNet [a] generalizes the charts idea even further which the paper does not cite. Therefore, I would suggest toning down statements that make it seem like this is a novel contribution such as "...which we call charts." (lines 116, 43) - While adopting AtlasNet for the problem, the paper completely ignores a key property of AtlasNet: universal approximation. AtlasNet is capable of deforming any (non-integer continuous) point on a UV chart to 3D. Because the current approach uses fixed vertices on the chart, it does not have this property. Therefore, it is closer to FoldingNet [b] than to AtlasNet in design. The lack of universal approximation is not a big limitation, but still it would be useful to mention these differences. - I was confused about how hand and object pose are estimated? It would seem that in order to align the touch charts and the vision charts registration is required. How is this obtained? This is not clear. - The touch signal outputs are quite close the the orthographic depth images predicted by the U-net (as shown in supp doc). I wonder if this network is needed in the first place? How much work does this network do? - Why are 3 iterations used? Was this determined empirically? [a]: Learning elementary structures for 3D shape generation and matching, 2019 [b]: Foldingnet: Point cloud auto-encoder via deep grid deformation, 2018 --------------------------- Thank you for the author response. I increased my score slightly to reflect the rebuttal. I still believe that the paper overclaims some contributions and should tone down and discuss relationship to FoldingNet, etc.

Correctness: Yes, except for the claim about how humans use touch for local and vision for global.

Clarity: The paper is mostly well-written. Justification of certain design choices (e.g., num of iterations) is not provided. Some details are lacking (e.g., how is hand pose relative to object acquired?).

Relation to Prior Work: There are several related pieces of work that are not cited. Here's a non-exhaustive list. - The presented approach is closer to FoldingNet than to AtlasNet. Foldingnet: Point cloud auto-encoder via deep grid deformation, CVPR 2018 - Learning joint reconstruction of hands and manipulated objects, Hasson et al. 2019 - Learning elementary structures for 3D shape generation and matching, 2019

Reproducibility: Yes

Additional Feedback: Overall, I really liked the idea of the paper. But I feel like the algorithmic components were not well motivated especially with respect to the adoption of graph networks in the AtlasNet chart context.


Review 2

Summary and Contributions: This paper presents a learning pipeline that fuses both visual and tactile signals for 3D obejct reconstruction, and shows that additional tactile signals from a few grasp points improve the accuracy of single image 3D reconstruction on ShapeNet. It also introduces a simulation dataset using a simulated tactile sensor and ShapeNet objects.

Strengths: This paper explores the idea of leveraging tactile signals to compensate for the shortcomings of image-based 3D reconstruction, particularly single-image reconstruction approches hindered by occlusions and scale ambiguity. To validate this hypothesis, the authors created a simulation dataset and designed a reconstruction pipeline with a chart-based mesh representation and an iterative refinement step for aggregating chart reconstructions from both visual and tactile observations. Particularly, the model uses graph convolutions on neighbor vertices and establishes connections between visual and tactile charts. This iterative pipeline replaces allows the model to be trained jointly, whereas previous methods adopt a multi-stage process. The results confirm that using the additional tactile signals consistently improve the single-image reconstruction accuracy.

Weaknesses: My major concern is with the simulation setup and the lack of comparison to previous visuotactile reconstruction methods. [Simulation] - The simulation dataset considered in this work is rather simple, with simple shapes, textures and accurate tactile signal with ground-truth pose from simulation. And the touch network is also pretrained (in line 123; with ground-truth depth?). It's not clear how big the gap between this simulation dataset and real tactile sensing is, and how much the performance improvement translates to real world scenarios. - Tactile signals are useful in two ways: (1) it provides some information about the rough 3D scale of the objects and occlusions, (2) it provides signal for local surface details. Most of the synthetic objects considered in the experiments consist of mainly smooth surfaces, which seem unfit for validating the second hypothesis. Does it make sense to add surface details onto the synthetic objects, and expect a more significant boost in reconstruction accuracy with the tactile signals? [Experiment] - In the touch-only experiment in Tab 2, is it using only one single grasp? If only a few contacts with the surface are available, ie, only very few local surface patches are observed, the model surely can only resort to the category-level prior shape and hence results in poor performance. Is this the point regarding the conclusion that tactile signals are local? My interpretation of tactile signals being local is that: if humans touch an object *extensively* with their eyes closed, they are still bad at reconstructing the overall shape of the object. This is partly due to the fact we do not have ground-truth (relative) 3D poses of the contacts, whereas this is not the case here where GT poses are given. - Why are previous methods on visuotactile reconstruction not compared?

Correctness: Yes.

Clarity: - I think the method sections could be better organized and structured. I find it quite difficult to digest the blocks of details and map them back onto the overall pipeline. - It's better to be consistent with terms in the method description and in the experiment description, eg sharing in exp. vs. exchange/communication in method.

Relation to Prior Work: Yes, but no comparison is made.

Reproducibility: Yes

Additional Feedback: ----- post rebuttal ----- I appreciate the authors' efforts in the rebuttal. Overall, I think this paper presents an interesting topic, and is overall a solid submission. Although the experiment setup is rather simple, I think it suffices to validate the claims in this proof-of-concept paper.


Review 3

Summary and Contributions: This paper proposes to fuse the touch and vision signal as the input for complete 3D shape reconstruction. The touch signal provides a close-to-accurate reconstruction of a small set of 3D shape charts. Then the vision signal is fused to complete the full shape using a learning based approach. The detailed experiments prove that the fusion of touch and visual signals has better performance. This paper also includes the detailed analysis of occlusion factors and the number of grasps.

Strengths: - The experiments are detailed and thorough. The vision based single view 3D reconstruction method achieves the state-of-the-art (shown in supplemental). The fusion with the touch signal further boosts the overall performance. There're also detailed analysis regarding the occlusion factors and the number of grasps. - The proposed chart based representation is a good fit to the task. - It's inspiring to see the fusion of interactions and vision in this paper.

Weaknesses: - The fusion of the touch and vision can be explored in a more elegant way. The current approach completes the 3D shape based on the vision signal conditioned on the already recovered shape from the touch signal. It would be interesting to see if the intermediate representations of the touch signal (e.g. point clouds, orthographic depth) can be a set of good features to help improve the vision based 3D reconstruction. - P.7, line 263, what's the "touch-only setting"? There is no clarification anywhere in the paper. The authors have addressed all my concerns in the rebuttal. I'm inclined to accept.

Correctness: Yes

Clarity: Yes

Relation to Prior Work: Yes

Reproducibility: Yes

Additional Feedback: See [Weaknesses]. I want to encourage further research regarding this research topic: reconstruction with interactions. Also I think the experiments are detailed and solid.


Review 4

Summary and Contributions: The presented paper proposes a novel multimodal approach to learn 3D shape reconstruction from vision and haptic signals. The main contributions of this work are that a novel architecture and reconstruction scheme is proposed to perform sensor fusion for the reconstruction of 3D shapes. Furthermore, the paper discusses a number of ablation studies as well as experiments with single and multi-touch feedback.

Strengths: - The addressed problem of reconstructing 3D shapes, with hand-object interaction, is novel and interesting. - The introduced sensor-fusion approach appears novel and provides a new way to reconstruct objects. - The paper is mostly easy to read and the technical design choices are sound and easy to comprehend.

Weaknesses: - A strong negative aspect of the paper is that it does not discuss any limitations or corner cases. As is, it is not clear in which situation the method fails. - Only a small set of objects was explored. - The method was only shown to work in a simulated environment.

Correctness: The method seems technically sound and the paper does not make unsupported claims.

Clarity: The paper is well-written and easy to follow.

Relation to Prior Work: Related work is discussed appropirately with no important work missing.

Reproducibility: Yes

Additional Feedback: - The caption of Figure 1 is difficult to comprehend: the caption describes what is proposed in the paper and now what is shown in the figure. The latter would be more helpful. - More details need to be provided for Equation (1): when I am not mistaken $v$ and $\mathcalN_v$ are not defined in Eq (1). - {C_i} is introduced after it is used, which makes sentences around L136 more difficult to understand than need be. - In Figure 3. the rightmost sub figure is not very meaningful. When I understand correctly it is supposed to depict a 3D shape part that was generated from the deformation of a single chart. If this is the case, why does it only have four faces and not 10 as shown in the left sub figure? Otherwise, if it is supposed to show a whole shape made of multiple charts, it is confusing to only show a tetrahedron as the resulting, reconstructed shape. - In Section 3.1. it is not clear why mesh charts need to communicate. It becomes clear later on in the text, but it would be helpful to clearly state this in the beginning of Section 3.1. - Furthermore, it is not clear why vertices in independent charts cannot communicate (Line 138-140). How is a neighborhood defined here? - In Figure 5 it is not clear what is meant by 'ortographic depth from touch reading image'. This should be described more closely to what is discussed in Section 3.2. - The touch signal images/illustrations shown across the various figures are not particularly meaningful: (1) they seem to always be the same and it is not clear why that would be the case; (2) it is not reported what the color coding is supposed to mean. - The discussion on defining depth maps and placing lights to define depth maps is not very clear and needs to be rewritten. Why is it necessary to render something with Phong for generating depth intensity values? - The paper should provide a discussion on the resolution of the generated shapes. The reconstruction quality seems to be highly dependent on the resolution of charts. How does it vary if charts have different resolutions? How is the iterative refinement scheme affected by the resolution? - Finally, it would be helpful to position the performance and quality of this work in the context of other -- single modality -- shape reconstruction methods.

[Author Response · NeurIPS 2020]

We thank all reviewers for their feedback and insightful comments. We are pleased to see our contributions warmly received: "It's inspiring to see the fusion of interactions and vision in this paper" (**R3**); "The addressed problem of reconstructing 3D shapes, with hand-object interaction, is novel and interesting" (**R4**); and "Overall, I really liked the idea of the paper" (**R1**). We will modify the paper according to reviewers' suggestions and we will introduce their comments with respect to the paper presentation.

**R1.** Antropomorphization of the motivation: We used human-inspired examples in the abstract and intro to motivate the complementarity of vision and touch when performing 3D object understanding. It was not our intention to antropomorphize the algorithm but rather to provide the task motivation. We will revisit the abstract and intro to clarify this. Model presentation: Our algorithm was built upon the use of graph networks for mesh deformation (e.g. [59] and [52]), due to their *strong performance* in 3D reconstruction tasks, and the *advantageous resolution properties* of meshes. Moreover, using charts enables the *disentanglement of visual and tactile information* and, as result, the model can enforce *touch consistency* in the final prediction. Hence, we introduced our algorithm as integrating charts within mesh-based approaches. Although we could have *alternatively* presented it as adding meshes to the AtlasNet charts, we disagree with the premise that one way of presenting is superior to the other. Nevertheless, we will revise the paper to motivate our approach from both perspectives. Charts: Atlases and charts are well known concepts in topology for describing manifolds and we did not intend to claim our use of them as a novel. We will change the presentation of the intro to reflect this, in particular, we will change "which we call charts" to "called charts [ref]". However, we do want to highlight that, to the best of our knowledge, our approach is the first to use these concepts for joint reconstruction from vision and touch signals. Universal approximation: As pointed out by **R1**, this is not a big limitation. However, the proof for universal approximation of AtlasNet relies on sampling and then passing sufficiently many points to an ideal MLP to epsilon-approximate a manifold. The FoldingNet decoder uses a single 2D grid of fixed points and transforms each point independently. By contrast, our model deforms the vertices of many charts by aggregating neighboring information at each layer of the graph network (see Eq. 1). We will add the suggested references. Pose information: The hand pose is available to the method for inference, and all charts are predicted within hand's reference frame. UNet role: The UNet model does currently model a non-invertible mapping due to smoothing in the rendering process. # iterations: 3 refinement iterations were used based on empirical validation, and backed by prior works [59, 52].
**R2.** Simulation simplicity: Our simulator is based on objects coming from a standard 3D reconstruction dataset - ShapeNet. While it is true that the objects are not highly complex (e.g. in terms of surfaces and scales), our dataset is the first of its type. In addition, despite its simplicity, the dataset was sufficient to clearly demonstrate the benefit of touch for 3D reconstruction. Touch information: In our setting, touch is useful for completely reducing the uncertainty of a surface's local position and structure. Adding surface details for 3D reconstruction would probably not lead to large performance variations as they amount to very little variation in overall shape.Touch-only experiment in Tab.2: Yes, in this experiment we report results for a single grasp. We observed that models leveraging touch only reach almost perfect reconstruction on touch site but poor global reconstruction quality, confirming that the benefits are local. Comparison to prior work: The comparison for a variety of visuotactile reconstruction baselines is reported in Tab. 1. We do not compare directly to prior work, as we are unaware of work directly applicable to our setting (see l. 93–104), though we would have been happy to compare to or discuss the relevance of other work, had **R2** highlighted them.
**R3.** Direct use of intermediate representations: We consistently found our approach to outperform models naively conditioned on intermediate touch representations, e.g. Tab. 1 (rows 9 and 11) where the reconstruction algorithm receives intermediate features from the touch signals directly and yet performs notably worse. We hypothesize that this happens due to information imbalance in vision vs. touch for the reconstruction task. Touch only setting: Touch signals are exclusively passed to the algorithm (no vision). Thanks for the pointer, we will resolve this ambiguity.
**R4.** Limitations: (1) Predicted charts sometimes poorly overlap creating a noisy boundary, as opposed to smoothly connecting (see Figure 8). (2) Charts are not forced to form a continuous, connected surface, and so from our qualitative evaluation $\sim 5\%$ of predictions possess a chart detached from the remaining surface. (3) Our algorithm's dependence on full 3D scene information, which is perhaps unrealistic for un-simulated scenarios. We will include this discussion in the paper. Vertex neighborhood/charts: Vertex neighborhood is defined as the set of other vertices which share an edge with it *within the same chart*. Given this neighborhood definition, charts *do not share edges*, and so we enable communication among charts as described in l. 140–152. Interpreting of touch signals: Touch signals often look similar as the finger might only touch the object lightly, or the object may be smooth, making the touch reading not human-interpretable. Note that additional touch readings are depicted in Fig. 2 in the appendix. Simulating touch: The Phong model is not used to compute the depth signal, but to render a touch sensor reading from the depth representation. We provide a complete description of this process in Section 1.5 of the appendix. Prediction resolution: Decreasing the resolution of charts (# faces) decreases the representation power of the method and so reduces performance, while increasing it boosts performance, though only up to a limit. This is a property common to all representations in 3D reconstruction. Single modality comparison: For the vision only single modality setting see Section 3.6 and Tab. 4 in the appendix. Simulation simplicity: See R2 answer. Fig. 3: It depicts a whole shape made of multiple charts. Thanks for pointing this out, we will resolve this ambiguity.

[Meta-Review · NeurIPS 2020]

This paper proposes to fuse vision and haptic information to reconstruct 3D shapes for robotic hand manipulation. The reconstruction is done by representing the objects as a collection of deformable meshes (defined as charts in the previously published AtlasNet paper). The merging of the vision and touch charts is done using graph convolutional networks, with local and cross-modality communication between charts. Experiments are conducted in simulation, on a new dataset designed by the authors, with known hand and object surface structure, and vision and touch inputs. After rebuttal, reviewers gave scores between 6 and 7. They praised the writing, the ideas and the relevance of the problem. Reviewers R2 and R4 had concerns about the simulated environment. Reviewer R1 is concerned about too strong biology-related claims, as well as comparisons with "FoldingNet: Point cloud auto-encoder via deep grid deformation" (CVPR, 2018). Based on these comments, I believe the paper should be accepted as a poster. Please note that in the final version, the authors should address the many requests for clarification that have been raised by the reviewers.